# Molecular Mechanisms of Breast Cancer Metastasis to the Lung: Clinical and Experimental Perspectives

**DOI:** 10.3390/ijms20092272

**Published:** 2019-05-08

**Authors:** Braeden Medeiros, Alison L. Allan

**Affiliations:** 1London Regional Cancer Program, London Health Sciences Centre, Department of Anatomy & Cell Biology, Western University, London, ON N6A 5W9, Canada; bmedeir4@uwo.ca; 2London Regional Cancer Program, London Health Sciences Centre, Departments of Anatomy & Cell Biology and Oncology, Western University, London, ON N6A 5W9, Canada

**Keywords:** breast cancer, lung metastasis, pre-metastatic niche, exosomes, tumor secreted factors, targeted therapies

## Abstract

Breast cancer is the most commonly diagnosed cancer in women worldwide, and >90% of breast cancer-related deaths are associated with metastasis. Breast cancer spreads preferentially to the lung, brain, bone and liver; termed organ tropism. Current treatment methods for metastatic breast cancer have been ineffective, compounded by the lack of early prognostic/predictive methods to determine which organs are most susceptible to developing metastases. A better understanding of the mechanisms that drive breast cancer metastasis is crucial for identifying novel biomarkers and therapeutic targets. Lung metastasis is of particular concern as it is associated with significant patient morbidity and a mortality rate of 60–70%. This review highlights the current understanding of breast cancer metastasis to the lung, including discussion of potential new treatment approaches for development.

## 1. Introduction

Globally breast cancer is the most common malignancy in women, and 626,679 deaths worldwide in 2018 were attributed to it [1]. In the past, breast cancer has been a higher burden in developed nations due to risk factors associated with lifestyle [1]. However, in developing nations the incidence rates of breast cancer have increased in recent years due to advancements in health infrastructure and the adoption of a ‘westernized’ lifestyle [1]. In Canada, 1 in 8 women will develop breast cancer over their lifetime while 1 in 31 will die from their disease [2]. Of the deaths caused by breast cancer, over 90% are attributed to metastasis-related complications [3]. Metastasis is a poorly understood process that begins with the detachment of tumor cells from the primary tumor and their intravasation into the blood stream [4]. These circulating tumor cells (CTCs) eventually arrest in the capillary beds of distant organs and extravasate through the vascular wall into the parenchyma, resulting in the generation of metastatic colonies in the secondary site [4].

Breast cancer has a tendency to target the bone, brain, liver and lung; known as organ tropism [5]. For breast cancer patients with metastases; 30–60% have lesions in the bone, 4–10% in the brain, 15–32% in the liver, and 21–32% in the lung [6]. Lung metastases in particular tend to occur within 5 years of initial breast cancer diagnosis and have a significant impact on patient morbidity and mortality. Physiologically, these metastases disrupt normal lung function, resulting in coughing, labored breathing, hemoptysis, and eventual death. Lung metastasis remains difficult to treat, with an estimated 60–70% of patients who die of breast cancer having lung metastasis [7]. For patients with metastases confined solely to the lung, the prognosis is exceedingly poor with a median survival of only 25 months [8]. This poor outcome is attributed to the limited number of treatment options associated with inoperable lesions [9].

The underlying mechanisms that dictate which organ(s) become colonized by breast cancer are complex and influenced by many factors, one of which is molecular subtype. First described by Perou et al. (2000), breast cancer can be subdivided into four main clinical subtypes on the basis of gene expression profiles and receptor status (estrogen receptor [ER], progesterone receptor [PR], human epidermal growth factor receptor 2 [HER2]) and proliferation status as assessed by Ki67 [10]. These clinical subtypes (in order of increasing aggressiveness) include: luminal A (ER^+^/PR^+^), luminal B (ER^+^/PR^+^/ HER2^−/+^/Ki67^+^), HER2 overexpressing (ER^−^/PR^−^/HER^+^) and basal-like/triple-negative (TN) (ER^−^/PR^−^/HER2^−^). While bone is the most common site for metastasis across all subtypes, TN breast cancer has the greatest tendency to metastasize to the lung; occurring in ~32% of patients compared to ~21% of luminal A/B and ~25% of HER2+ patients [6]. However, the timing and mechanisms by which breast cancer molecular subtype may influence metastasis to the lung is not yet understood.

In this review, we summarize current advancements in the understanding of molecular mechanisms that drive breast cancer metastasis to the lung. By integrating the complex body of work that surrounds this topic, we highlight key therapeutic targets and potential/emerging treatment approaches.

## 2. The Lung Metastatic Niche

The process of metastasis is highly inefficient, with less than 0.01% of primary tumor cells successfully completing the metastatic cascade to develop macrometastases at the secondary site [11]. Clinically established patterns of organ-specific metastasis suggest that the site in which the cancer grows successfully is not random, but rather influenced by the microenvironment in the secondary organ. This phenomenon was first described by Stephen Paget in 1889, who hypothesized that cancer cells (the “seed”) grew preferentially in the microenvironment of select organs (the “soil”) only if the conditions at that site were permissive for growth [12]. Supporting this theory, our research group has shown that in the presence of organ-conditioned media from common sites of breast cancer metastasis (lymph node, lung, liver, bone, brain), breast cancer cells demonstrate organ-specific responses in proliferation and migration; indicating certain organs produce soluble components that support metastatic behavior [13]. However, in the early 1900s, James Ewing suggested a competing theory that organ-specific metastasis was regulated solely by physiological blood flow patterns [14]. Certainly the physical characteristics of organs such as the lung lends weight to Ewing’s theory, particularly for breast cancer. The lung is the first major capillary bed that a breast cancer cell encounters after escaping into the bloodstream. As tumor cells circulate through the lung, they may come into contact with as much as 100 m^2^ of surface vasculature. Since these tumor cells are approximately five times larger than the exceedingly narrow pulmonary capillaries, the likelihood of breast cancer cell arrest in these capillary beds and subsequent extravasation into the lung tissue is high [15,16]. The lung capillaries are comprised of endothelial cells that are encapsulated by a basement membrane and adjacent alveolar cells. To facilitate transendothelial migration and extravasation, tumor must express cell surface markers specific for the lung microenvironment [15,16]. However, although extravasation may occur fairly easily via these physical processes, the ability of individual metastatic cells to successfully transition to micrometastases and subsequently progress to macrometasases is quite rare, and thus these final events represent rate-limiting steps in metastasis that rely on the optimal collaboration of “seed” and “soil”. Therefore, it is likely that Paget’s and Ewing’s theories are highly complementary in the development of metastasis to the lung [17].

More recently, growing evidence that the primary tumor has the potential to “prime” or augment distant organ microenvironments in preparation for metastasis has added a further level of complexity to Paget’s seed and soil theory [18,19,20,21]. The generation of this “pre-metastatic niche” is hypothesized to be critical for the process of metastasis, and can be divided into four phases consisting of priming, licensing, initiation and progression [22]. Priming is initiated by the secretion of tumor-derived soluble factors (TDSFs) and/or exosomes by the primary tumor as it undergoes uncontrolled proliferation and becomes hypoxic. These molecules target the bone marrow for recruitment and initiate the remodeling of the target secondary organ, generating a pre-mature metastatic niche. Bone marrow-derived cells (BMDCs) and immune regulatory/suppressive cells are then progressively recruited to the secondary site by the continual secretion of factors from the primary tumor. These processes facilitate the licensing phase, generating an immune-suppressed environment and an extracellular matrix (ECM) conducive for cancer colonization. Disseminated cancer cells that enter this fertile metastatic niche may stay in a dormant state until the conditions at the secondary site can support tumor outgrowth into micrometastases; termed initiation. In the final progression stage, the growth of micrometastases is regulated by tumor secreted factors and other regulatory cells that infiltrate the secondary site, thus enabling transition to macrometastases [22].

In order for these processes to occur successfully, there must be a well-choreographed sequence of molecular and cellular events that enable the generation of a fertile pre-metastatic niche, and this is regulated by a variety of tumor-secreted factors, exosomes and stromal components. In particular, the close interplay between primary tumors and lung priming was first highlighted by Lyden and colleagues (2005). They demonstrated that bone marrow derived hematopoietic progenitor cells (HPCs) expressing vascular endothelial growth factor receptor-1 (VEGFR1) and very late antigen–4 (VLA-4) targeted areas of the lung with increased fibronectin deposition [23]. Upon binding, VLA-4^+^VEGFR1^+^ HPCs secrete MMP9 to induce pro-metastatic changes in the lung ECM [23]. Subsequent studies have elucidated how the BMDCs are recruited to the pre-metastatic niche and the underlying mechanisms of increased fibronectin deposition in the lung; believed to be regulated by tumor-derived exosomes and a variety of tumor-secreted and stromal-derived factors. These are summarized in Table 1 and Table 2 and described in greater detail below.

## 3. Tumor-Derived Exosomes

Tumor-derived extracellular vesicles can be subdivided on the basis of size including apoptotic bodies (1000–5000 nm), microvesicles (200–1000 nm) and exosomes (30–150 nm) [47]. Exosomes are formed through the endosomal pathway and predominantly released from cells through fusion with the plasma membrane [48,49]. Tumor-derived exosomes (TDEs) have been shown to play a significant role in modifying the lung microenvironment. This is highlighted by recent studies involving pre-treatment of mouse models with TDEs from lung-seeking breast cancer cell lines that showed the potential to “educate” the lung, inducing changes that made it more susceptible to metastasis [50]. Exosomes secreted by the primary tumor have the ability to target the lung through use of integrins such as ITGα_6_β_1_ [27]. Upon targeting specific organs, breast cancer-derived exosomes can deliver their cargo of RNA, DNA and proteins to induce pro-metastatic changes in the lung [51]. Exosome production and packaging is not static but is instead regulated by several factors including environmental stimuli, such as hypoxia [52,53]. It has been demonstrated that breast cancer exosome production is increased substantially in hypoxic conditions in a HIF-1α dependent manner [54]. Furthermore, exosomes have the potential to translate properties such as chemotherapy resistance and increased invasiveness to recipient breast cancer cells [55,56].

### 3.1. Exosomes and Immune Suppression

The process of generating the pre-metastatic niche in lung is highly reliant on immune-suppression to ensure that CD8^+^ T cells, natural killer (NK) cells and patrolling monocytes are masked from the presence of tumor cells trying to establish themselves as metastatic lesions in the lung [57,58]. Interestingly, Yang and colleagues (2018) demonstrated that breast cancer-derived exosomes expressing programmed cell death-1 (PD-L1) on their surface have the ability to blunt T-cell activation and killing activities, effectively protecting tumor cells from immune surveillance. By stunting the immune response, tumor cells have the potential to successfully seed and colonize distant organ sites, such as the lung [24]. While cancer cells in the bloodstream try to avoid circulating immune elements, there is growing evidence to suggest that there is a connection between immune cell dysregulation and chronic inflammation at the pre-metastatic site [59,60,61]. Macrophages (key regulators of the immune response and part of the innate immune system) phagocytose invading cells in order to induce the expression of cytokines and chemokines [62]. Chow et al. (2014) demonstrated that exosomes derived from MDA-MB-231 and MCF7 breast cancer cell lines have the ability to hijack lung macrophage activity by activating the NF-κB pathway, resulting in the expression of the pro-inflammatory markers IL-6, TNFα, G-CSF and CCL2 and promoting lung metastasis in vivo [63].

Precipitated by the work of Lyden and colleagues, BMDCs have emerged as a significant contributor to establishing a pre-metastatic niche [64,65,66]. Furthermore, although it is well-established that myeloid-derived suppressor cells (MDSCs) enable tumor progression, their development during tumor growth was unknown [67,68,69,70]. Xiang and colleagues (2009) demonstrated that bone marrow myeloid cells can be forced to differentiate into myeloid-derived suppressor cells (MDSCs, CD11b^+^Gr^−^11^+^) by breast cancer derived exosomes [71]. The resulting change induces the accumulation of MDSCs expressing Cox2, IL-6, VEGF and arginase-1 at the lung, generating a pro-inflammatory and immune suppressed environment permissive for metastasis [71]. In a similar study, Peinado et al. (2012) demonstrated that exosomes from highly metastatic melanoma cells had the ability to “educate” BMDCs from non-tumor bearing mice, pushing towards a pro-vasculogenic and pro-metastatic phenotype via the upregulation of MET [72]. The relevance of this process to breast cancer lung metastasis has yet to be investigated.

### 3.2. Exosomes and Stromal Cells

Beyond interactions with immune cells, breast cancer-derived exosomes have the ability to influence the status of the lung microenvironment by modulating the function of stromal cells. Fong et al. (2015) demonstrated that exosomes isolated from the MDA-MB-231 breast cancer cell line contained miR-122, and once applied to lung fibroblasts these exosomes were able to reprogram glucose metabolism, reducing glucose uptake by inhibiting pyruvate dehydrogenase activity [25]. This suggests that prior to colonization of the lung, secreted exosomes from the primary tumor can reduce glucose uptake, allowing for newly arrived cancer cells to have enough energy to facilitate rapid proliferation. Complementary to this data, Zhou and colleagues (2014) demonstrated that exosomes released from MD-MB-231 cells were also enriched with miR-105 [26]. Tail vein injection of miR-105 containing exosomes resulted in modulation of the vasculature of common sites of metastasis such as the lung such that it became “leaky” [26]. It was determined that exosomal delivery of miR-105 to endothelial cells resulted in the downregulation of the tight junction protein ZO-1 [26]. Taken together, these studies provide further evidence to support the concept that breast cancer derived exosomes play a critical role in establishing a permissive niche in the lung required for metastatic colonization, and highlights the possibility of considering exosomes in the clinical setting.

### 3.3. Exosomes as Clinical Biomarkers

Tumors secrete a number of factors into peripheral circulation (Table 1) that have the potential to serve as a method of clinical monitoring of disease progression. Increased attention has thus been put towards developing non-invasive blood-based biomarker approaches [73,74,75]. Many current methods have focused on enumeration and characterization of circulating tumor cells (CTCs), but this has proved to be a difficult task due to their sparse concentrations in blood [76,77]. In patients with early stage breast cancer, the CTC detection rate ranges between 23–37%, and currently no effective strategy exists to leverage CTCs analysis in order to predict which organs might be affected by metastasis [78]. In comparison, tumor-derived exosomes may provide an attractive alternative as they are stable in blood, have the ability to be isolated from most bodily fluids (blood, urine, semen, milk) and are present in circulation at similar quantities as soluble proteins (10^5^·mL^−1^) [79,80]. The innate issue with exosomes is differentiating their origin from normal or cancerous tissue. Etayash and colleagues (2016) demonstrated that by utilizing the characteristics of tumor-derived exosomes (such as the overexpression of CD24, CD63 and EGFR), they were able to isolate exosomes of breast cancer origin by a multiplexed cantilever array sensor [81]. By isolating breast cancer exosomes, proteomic analysis may provide the opportunity to identify sites susceptible to metastasis by identifying the presence of specific organotropic integrins. Additionally, RNA analysis coupled with such proteomic data may provide insight into how the exosomes are altering the secondary site, providing a targeted approach to circumvent potential metastases. An alternative method developed by Zhai et al. (2018) demonstrated that isolating patient plasma and incubating the sample with Au nanoflare probes specific to the pro-metastatic exosomal miR-1246 was able to identify 100% of breast cancer patients with metastatic disease [82]. These methods of early detection have great promise but must be refined, validated in the clinical setting, and ideally coupled with new treatment approaches to have clinical applicability.

### 3.4. Therapeutic Implications of Exosomes

Beyond biomarkers, exosomes could provide an effective strategy to treat inoperable metastatic lung lesions. In the majority of cases, lung metastases are multidrug resistant and efforts to change treatments is thwarted by most chemotherapeutics having low aqueous solubility requiring alternative delivery methods [83,84,85]. Exosomes, compared to other proposed delivery methods, have the potential for organ-specificity and privileged immune status that results in reduced drug clearance [86]. Applying this, Kim et al. (2016) demonstrated that exosomes released by macrophages can be loaded with paclitaxel using ultrasound treatment and used to induce cytotoxicity in multidrug resistant lung cancer cells [87]. Future work in the context of breast cancer lung metastasis is crucial to move towards translation of this potential therapeutic approach to the clinic.

Interestingly, while tumor-secreted vesicles provide hope for earlier detection of lung metastasis and treatment, there is emerging evidence to suggest that they may actually further complicate treatment outcomes. Keklikoglou et al. (2019) recently demonstrated that the administration of taxanes and anthracyclines to mice bearing breast tumors induced the release of extracellular vesicles with an increased pro-metastatic capability [88]. These extracellular vesicles contained elevated amounts of annexin A6 which targets lung endothelial cells to induce NF-κB activation, resulting in CCL2 release that caused Ly6C^+^CCR2^+^ expansion at the lung that enables the establishment of a fertile pre-metastatic niche [88].

In addition to chemotherapy, immunotherapy has emerged as a revolutionary approach to cancer treatment and management. Several recent clinical trials have focused on determining the efficacy of PD-L1 inhibitors in the context of metastatic breast cancer [89,90]. Interestingly, in metastatic melanoma, Chen et al. (2018) were able to demonstrate that the amount of PD-L1 expressed on tumor-derived exosomes was a predictor for response to anti-PD-L1 therapy [91], whereby responders had lower baseline levels of PD-L1 expressed on exosomes, and after 3–6 weeks of treatment the expression was more pronounced for responders [91]. These observations indicate the importance of considering the role of exosomes when designing treatment regimens for patients with metastatic disease, from the perspective of drug delivery and response.

## 4. Tumor-Derived Secreted Factors

In addition to the secretion of exosomes, breast cancer primary tumors release a variety of other factors that have the potential to prime or augment the lung microenvironment, known as tumor derived secreted factors (TDSFs) (Table 1). An aspect of the niche that is critical for successful metastatic colonization is the status of the ECM. In a pro-metastatic state, secondary organs such as the lung upregulate the expression of several ECM components including versican, tenascin-c, periostin and fibronectin [37,38,39,40,41,92]. As with exosome production, the effect that primary tumor-secreted factors have on the secondary site is regulated by both environmental stimuli and interactions with stromal cells that comprise the tumor microenvironment. Hypoxia within the primary tumor influences a wide variety of processes, including the increased expression of lysyl oxidase (LOX). LOX is an amine oxidase that crosslinks collagen and elastins in the ECM and is associated with a variety of pro-metastatic processes [42]. Erler et al. (2009) demonstrated that under hypoxic conditions breast cancer primary tumors increase LOX expression, inducing crosslinking of collagen in the lung [34]. This change in the ECM enables the adhesion of CD11b^+^ myeloid cells, and upon binding produces MMP2 resulting in the cleavage of collagen that is required for recruitment of BMDCs and cancer cells to the lung microenvironment [34]. This highlights an interesting observation that the status of the ECM determines which cells are recruited to the lung and can be changed by these recruited cells to promote metastatic seeding.

## 5. Stromal-Derived Influences

In addition to hypoxia, the primary tumor may be influenced by cells that comprise the tumor microenvironment which includes immune cells, endothelial cells, adipocytes and several other cell types [93] that produce factors that influence the metastatic process (Table 2). Emerging evidence suggests that a subset of activated stromal cells termed cancer-associated fibroblasts (CAFs) cause a multitude of changes in the behavior of breast cancer cells [94,95,96]. These CAFs are a heterogenous population of cells that vary in origin and are characterized by the expression of several markers including αSMA and PDGFR-β [97]. In the context of breast cancer, CAFs drive the progression of metastasis by paracrine signaling and mechanical pressure on the cancer tissue [98]. In relation to paracrine signaling, CAFs in the tumor microenvironment have been shown to secrete IL-32 and, once bound to integrin β3 on the cell surface of breast cancer cells, results in the activation of p38 MAPK signaling that causes increased expression of EMT markers such as fibronectin, N-cadherin and vimentin [99]. In order to determine the influence of CAFs on metastasis in vivo, BT549 breast cancer cells were co-cultured with CAFs and injected subcutaneously in mice, resulting in increased lung metastases [46].

While the primary tumor has the ability to influence the immune response directly, this is also achieved by the help of lung stromal cells. Breast primary tumors have the ability to up-regulate the expression of S100A8 and S100A9 in the lung microenvironment [46]. These proteins are part of the S100 family which are characterized as calcium-binding cytosolic proteins. The expression of these elements act as strong chemoattractants for both neutrophils and macrophages and promote the metastatic potential of the breast primary tumors [46]. Specifically in the context of lung metastasis, S100A8 and S100A9 have been shown to induce the recruitment of Mac-1^+^ myeloid cells to the lung that results in the secretion of migration stimulating factors (TNFα, MIP2 and TGF) and ECM remodeling [100].

The primary tumor must influence several molecular and cellular processes in order to ensure that metastasis to the lung is successful. One important example of this is the interplay between BMDCs and the primary tumor, a relationship first highlighted by Lyden and colleagues [23] that has gained increasing traction in recent years. Interestingly, previous research has demonstrated that bone marrow derived mesenchymal stromal cells (MSCs) have the potential to differentiate into CAFs upon co-culture with tumor cells [101,102]. Building off this observation, Raz et al. (2018) used collagen-α1 tracking of transplanted bone marrow cells to demonstrate that MSCs make up a substantial proportion of CAFs that are present in the breast primary tumor and lung lesions [103]. These MSC-derived CAFs generate a unique inflammatory profile depending on the site they are recruited to and promote pro-metastatic features such as angiogenesis, enabling breast cancer metastasis to the lung [103]. Interestingly, the interaction between breast cancer cells and bone marrow cells is also crucial for cancer dormancy. Breast cancer dormancy is a significant clinical concern, as cancer cells in this state remain in mitotic arrest, making them resistant to drugs targeting highly proliferative cells. Upon recurrence, these senescent cells have the potential to resurge as progressive metastatic disease. It is widely accepted that the bone marrow serves as a sanctuary site for dormant breast cancer cells, and although this process is believed to be mediated by MSCs, the molecular underpinning of this relationship remains poorly understood [104,105,106,107,108]. Expanding on this interaction using a transwell model to recapitulate communication between MSCs and breast cancer cells, Bliss et al. (2016) showed that breast cancer cells prime MSCs to secrete exosomes containing miR-222/223 which induce senescence in the recipient cancer cells [109].

To recruit BMDCs to the lung, previous literature has demonstrated that the state of the secondary site must be augmented to promote adhesion. Angiogenic factors such as VEGF released from lung-seeking breast cancer cells can activate the Src-FAK pathway in lung endothelial cells, resulting in increased expression of lung adhesion molecules and enhanced integrity of vascular permeability. Lung endothelial cells also express PGE2 in response to VEGF, which acts as a powerful chemoattractant to recruit BMDCs and tumor cells to the lung [110]. Under hypoxic conditions the expression profile of the primary tumor changes, causing variations in downstream targets. Previous literature has demonstrated that HIF-1 induces increased expression of Carbonin Anyhydrase IX (CAIX) which has been shown to be required for breast cancer metastasis [35,111,112,113]. Taking this information, Chafe et al. (2015) were able to demonstrate that CAIX is required for the production of G-CSF by breast cancer cells, which in turn is responsible for the recruitment of granulocytic myeloid-derived suppressor cells to the lung for generation of a lung pre-metastatic niche [114].

## 6. Potential for Clinical Translation

Successful prevention of breast cancer metastasis to the lung will be dependent on identifying and treating not only the key characteristics of the “seed” (lung-seeking cancer cells), but also the “soil”; including the lung microenvironment and the pre-metastatic niche. In particular, the lung pre-metastatic niche has the potential to have significant implications in determining the risk that a particular patient may have for developing lung metastases. However, thus far most studies pertaining to the pre-metastatic niche have been limited to mouse models, although this is beginning to evolve towards the clinic. For example, knowledge that S100A8 and S100A9 are crucial for the generation of the pre-metastatic niche in the lung led to the development of S100A9 specific single photon emission computed tomography (SPECT) whole body imaging which has been tested in a pre-clinical breast cancer metastasis model [31]. In addition, using the knowledge that exosomes have the propensity to target specific organs, Nikolopoulou and colleagues (2016) isolated exosomes from the breast cancer cell line 4175-LuT which has a propensity to metastasize to the lung. The isolated exosomes were then labeled and injected into tumor-naïve female nude mice. The tissues from the mice were harvested, indicating a high accumulation of exosomes in the lung [115]. This method opens the potential to use high resolution, non-invasive imaging such as SPECT to identify pre-metastatic niche formation. Similar to this work, Soodgupta, et al. (2013) targeted VLA-4 expressed on BMDCs localized to the lung with a radiopharmaceutical. PET was subsequently performed to provide an effective method of detecting BMDCs in the pre-metastatic niche [116].

Upon identifying the presence of a pre-metastatic niche in the lung, the next step would involve preventing lung metastasis. To date, potential therapeutics targeting the pre-metastatic niche have mostly been assessed in the pre-clinical setting. However, the LSD1-specific inhibitor INCB059872 which is currently at phase 1 clinical trial for relapse of Ewing Sarcoma has been shown to reshape the myeloid compartment in a spontaneous lung metastasis model. Lee et al. (2018) demonstrated that INCB059872 reduced the migration of TN breast cancer cells, significantly reduced MSDC infiltration of the primary tumor and lung associated with a reduction in circulating CCL2, corresponding to a decrease in metastatic lung foci [117]. Alternatively, myeloid cells can be recruited to the lung by exosomes containing CSF-1 which are produced under hypoxic conditions. CSF-1 is associated with myeloid cell survival, proliferation and differentiation and has been previously been shown to be inhibited by GW2580. Pretreatment of mice with GW2580 prior to tumor implantation resulted in a significant decrease in myeloid cell recruitment to the lung and increase in anti-tumorigenic M1-macrophages [118]. Finally, as with exosomes, hypoxia has the potential to promote the generation of a pre-metastatic niche in the lung. Under hypoxic conditions hypoxia-inducible factors (HIFs) become activated, resulting in the induction of LOX and lysyl oxidase like (LOXL) proteins which are responsible for collagen remodeling and BMDCs recruitment [119,120,121]. Wong et al. (2012) demonstrated that administration of two chemically different HIF inhibitors (digoxin and acriflavine) could prevent lung metastasis in an orthotropic breast cancer model. This inhibition was attributed to stunted LOX and LOXL expression which ultimately prevented collagen remodeling and BMDC recruitment [122].

Collectively, the results of these pre-clinical studies suggest that the effects of the pre-metastatic niche have the potential to be attenuated, preventing colonization. While these results are promising, the clinical benefit of these approaches for patients with an increased risk of lung metastasis has yet to be delineated. Efforts to elucidate the complicated underlying mechanisms that drive lung metastasis and pre-metastatic niche formation in the clinical setting have been hindered due to a limited number of patient samples derived from metastatic lesions. However, the implementation of rapid autopsy programs has the potential to provide these necessary samples [123]. In particular, the development of robust biobanks of matched primary tumors and metastatic lesions (from lung and other organs) would provide important research tools to advance this field. Profiling these samples could provide insightful information into the underlying mechanisms that drive metastasis to specific organs, uncovering potential biomarkers that could help predict and/or prevent organ-specific metastasis. Further research should also be focused on determining the length of time in which the metastatic niche within the lung lasts post-surgery, radiation and/or chemotherapy. This will allow for personalized treatment regimens that will ensure that even after the eradication of cancer, the recurrence of metastatic disease in the lung will be reduced or prevented altogether.

## 7. Conclusions

Breast cancer remains a significant burden in modern society, requiring further research to understand the underlying mechanisms that drive metastasis and how to target it. Metastasis to the lung is of particular concern as it is associated with high patient morbidity and mortality with no current effective strategies for early detection or eradication. Colonization of the lung is facilitated by a complex web of interactions with the tumor microenvironment, lung stroma, immune cells and BMDCs; and crosstalk between these components is mediated by exosomes and tumor/stroma-derived factors (summarized in Figure 1). These secreted elements are dynamic and vary based on environmental stimuli and interaction with stromal cells that infiltrate the tumor microenvironment and secondary site. Together these interactions transition the lung microenvironment into a fertile niche susceptible for cancer cell colonization. The therapeutic approaches described above hold promise for preventing lung metastasis, but have currently only been investigated in a pre-clinical setting, highlighting the need for further development and research in this area. Concurrently, a major limitation in this area of research is the lack of clinically relevant biomarkers to determine if a patient has an increased risk for lung metastasis. To address this, further research must focus on the use of exosomes as a predictor of organ-specific metastasis including validation of this in patient samples. These gaps in our current understanding demonstrate the complexity of metastasis, indicating that to have a significant impact on this disease we must consider the consolidated effect of all factors together rather than just investigating them individually. In summary, developing a deeper understanding of the processes that enable breast cancer metastasis to the lung will lead to the development of therapeutic targets and biomarkers with the ultimate goal of preventing metastatic disease.

## Figures and Tables

**Figure 1 ijms-20-02272-f001:**
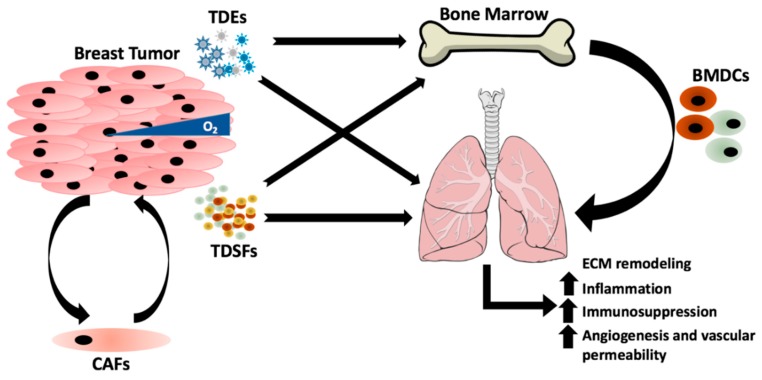
The underlying mechanisms that dictate the organ colonized by breast cancer are complex and influenced by many factors. Breast primary tumors regulate and prime the lung for metastasis by the secretion of tumor-derived exosomes (TDEs) and tumor-derived secreted factors (TDSFs), which target the bone marrow for recruitment of bone marrow-derived cells (BMDCs) to lung in order to induce changes in the extracellular matrix (ECM) that are conducive for metastasis. Release of TDSFs from the primary tumor are often regulated by stromal cells that compose the tumor microenvironment including cancer associated fibroblasts (CAFs) or environmental stimuli such as hypoxia.

**Table 1 ijms-20-02272-t001:** Molecular and cellular components secreted by breast cancer primary tumors that are associated with the promotion of lung metastasis.

Secreted Component	Molecule	Mobilized/Target Cell Type	Mechanism(s)	Reference(s)
Tumor-Derived Exosomes (TDEs)	PD-L1	T cells	Blunts T-cell activation and killing activities	[24]
	miR-122	Fibroblasts	Reprograms metabolic activity, resulting in decreased glucose need at pre-metastatic site by inhibiting pyruvate kinase	[25]
	miR-105	Endothelial cells	Uptake reduces expression of the gap junction protein ZO-1, promoting metastasis at the pre-metastatic site	[26]
	ITGα_6_β_1_	Lung SPC^+^ Epithelial Cells/ Lung S100A4^+^ Fibroblasts	Targets exosomes to lung to induce pre-metastatic niche formation	[27]
Tumor-Derived Soluble Factors (TDSFs)	P2Y2R	CD11b^+^ BMDCs	Mediates LOX expression, causing collagen cross linking in the lung recruiting CD11b^+^ BMDCs	[28]
	TGFβ	Cancer cells	Primes breast cancer cells by inducing ANGPTL4 which disrupts endothelial tight junctions at distant sites	[29]
	VCAM1	Endothelial cells	Facilitates transendothelial migration of tumor cells into the lung	[30]
	CSF-1	Macrophages	Recruits macrophages to the primary tumor, inducing an aggressive phenotype with a propensity to metastasize to the lung	[31]
	CXCR4/ CCR7	SDF-1/CCL21+ endothelial cells	Enables tumor cell adhesion to lung endothelium	[32,33]
	LOX		Leads to collagen crosslinking and recruitment of CD11b^+^ BMDCs	[34]

PD-L1, Programmed Death—Ligand 1; miR, microRNA; ITGα_6_β_1_, Integrin alpha 6 beta 1; SPC, Surfactant Protein C; P2Y2R, Purinergic Receptor; TGFβ, Transforming Growth Factor beta; ANGPTL4, Angiopoiten-like 4; VCAM1, Vascular Cell Adhesion Molecule 1; CSF-1, Colony Stimulating Factor-1; CXCR4, C-X-C Chemokine Receptor 4; CCR7, C-C Chemokine Receptor Type 7; SDF-1, Stromal Cell Derived Factor-1; CCL21, C-C Motif Chemokine Ligand 21; LOX, Lysyl Oxidase.

**Table 2 ijms-20-02272-t002:** Molecular and cellular components secreted by stromal cells associated with promoting breast cancer metastasis to the lung.

Secreted Component	Molecules	Mobilized/Target Cell Type	Mechanism(s)	Reference(s)
Stromal-Derived Factors (SDFs)	PGE2	BMDCs/Cancer Cells	Recruits BMDCs to the lung and enhances CTC adhesion	[35]
	ANG-2	CCR2^+^Tie2^−^ Macrophages	Recruits macrophages, which cause endothelial cells to release proinflammatory and angiogenic factors	[36]
	FN, TN-C, PSTN, VCAN	BMDCs and cancer cells	Promotes the adhesion of BMDCs and CTCs	[37,38,39,40,41,42]
	CCL2	Cancer cells	Produced by CCR2^+^ inflammatory monocytes, increases vasculature permeability	[43]
	MMP2	BMDCs	Remodels lung ECM	[44]
	HSF1	CAFs/cancer cells	Reprograms CAFs and cancer cells to promote metastasis into the niche	[45]
	Id3	VEGFR1+ BMDCS	BMDC-derived, required for recruitment of VEGFR1^+^ BMDCs to areas of increased fibronectin deposition in the lung	[23]
	IL-32	Cancer cells	CAF-derived, increases the metastatic potential of breast cancer cells	[46]

PGE2, Prostaglandin E2; ANG-2, Angiopoietin-2; CCR2, C-C Chemokine Receptor Type 2; FN, Fibronectin; TN-C, Tenascin-C; PSTN, Periostin; VCAN, Versican; BMDCs, Bone Marrow-Derived Cells; CCL2, C-C Motif Chemokine Ligand 2; CCR2, C-C Chemokine Receptor 2; MMP2, Matrix Metallopeptidase 2; ECM, Extracellular Matrix; HSF1, Heat Shock Factor 1; Id3, Inhibitor of Differentiation 3; VEGFR1, Vascular Endothelial Growth Factor Receptor 1; IL-32, Interleukin-32; CAF, Cancer Associated Fibroblast.

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
