# Peer review of "Molecular Mechanisms of Breast Cancer Metastasis to the Lung: Clinical and Experimental Perspectives"

_ijms, 2019, doi:10.3390/ijms20092272_

Reviewer 1 Report

This manuscript is well writen and quite informative. It described the breast to lung cancer metastatsis from multiple aspects. The reviewer has some minor suggestions.

The authors may want to describe or discuss the four types of breast cancers with regard to the likelihood of breast to lung metastasis.

The authors may want to discuss the preference of metastatic destination (lung vs. other organs) and the causes. 

Author Response

1. “This manuscript is well written and quite informative. It described the breast to lung cancer metastasis from multiple aspects.”
We thank the reviewer for this positive comment.
2. “The authors may want to describe or discuss the four types of breast cancers with regard to the likelihood of breast to lung metastasis.”
We thank the reviewer for this suggestion. We have now described the four subtypes of breast cancer with regard to the clinical incidence of lung metastasis (lines 56-58 in the revised manuscript).
3. “The authors may want to discuss the preference of metastatic destination (lung vs. other organs) and the causes.”
We have now added the overall prevalence of breast cancer metastasis to other organs versus lung (lines 40-41 in the revised manuscript). For the purposes of this review we decided to focus specifically on breast cancer metastasis to the lung, highlighting the physiological and lung microenvironment causes/characteristics that promote metastasis to this organ.

Reviewer 2 Report

This review by Drs. Medeiros and Allan presents a compilation of various molecular mechanisms/pathways that regulate metastasis of the breast cancers to lungs. The cancer-induced organ tropism is often the major cause of failure of treatment modalities, emergence of drug-resistant phenotypes, and consequent patient morbidity and mortality. Here the authors focus on human breast cancers and highlight various phenomena and signaling mechanisms by tumors and stroma that facilitate migration, localization, and growth of metastatic cells to the lungs. Authors provide a detailed account of tumor-derived exosomes and their molecular cargoes that facilitate communications between the tumor and stroma at the primary and metastatic sites, and highlight the potential of such tumor-associated/derived exosomal molecules as biomarkers for clinical use. Additional discussion is provided regarding the roles of the signaling and cellular components of stroma that often influence and facilitate the complex processes of distant metastases. Although the review focuses on the lung metastasis mechanisms by breast cancer cells, many of the signaling molecules and pathways could in principle also regulate metastases of other cancers to distant sites, particularly since many cancers often target nutrient-rich bone marrow for metastatic colonization. Overall, this review is well-written, easy to read, and deserving of publication in International Journal of Molecular Sciences for benefit of a broader readership within the cancer biology community. My minor editorial suggestions are as below:

1.       Please revise line 72 for better comprehension.

2.       Line 77, single metastatic cell.

Line 360, been previously shown to be.

Author Response

1. “Overall, this review is well-written, easy to read, and deserving of publication in International Journal of Molecular Sciences for benefit of a broader readership within the cancer biology community.”
We thank the reviewer for this positive comment.
2. “Please revise line 72 for better comprehension. Line 77, single metastatic cell. Line 360, been previously shown to be.”
These changes have now been made (lines 79, 88, and 375 in the revised manuscript).

Reviewer 3 Report

The authors present a wonderful review of breast cancer lung metastasis and detail several mechanisms and the current understanding of this highly specific metastatic process along with a nice historical context of the metastasis process. The comments below are intended to improve the manuscript.

1)      The introduction is a good primer for the topics to discussed later in the review. However, the rationale to focus solely on lung metastasis could be improved with more details on the prevalence and patient outcomes.

2)      The body of the review detailing mechanisms of lung metastasis primarily focus on physiological drivers of lung metastasis (i.e. seed and soil mechanisms). However, as the author point out early in the review, there is also the potential for physical attributes causing tropism (i.e. blood flow or metastatic tissue proximity to primary tumor). A more complete explanation of any of these physical factors on lung-specific metastasis from breast cancer would improve the manuscript.

3)      The conclusion could benefit from describing some tools needed to advance this field. One obvious things comes to mind in a large set of matched primary and metastatic tumors, both to lung and elsewhere. This type of tool would allow for answering the authors points of biomarkers that are lung-exclusive.

Author Response

1. “The authors present a wonderful review of breast cancer lung metastasis and detail several mechanisms and the current understanding of this highly specific metastatic process along with a nice historical context of the metastasis process.”
We thank the reviewer for this positive comment.
2. “The introduction is a good primer for the topics to discussed later in the review. However, the rationale to focus solely on lung metastasis could be improved with more details on the prevalence and patient outcomes.”
We appreciate this suggestion and have now added further details on prevalence and patient outcomes (lines 40-48 in the revised manuscript).
3. “The body of the review detailing mechanisms of lung metastasis primarily focus on physiological drivers of lung metastasis (i.e. seed and soil mechanisms). However, as the author point out early in the review, there is also the potential for physical attributes causing tropism (i.e. blood flow or metastatic tissue proximity to primary tumor). A more complete explanation of any of these physical factors on lung-specific metastasis from
breast cancer would improve the manuscript.”
We completely agree with the reviewer and have added some additional discussion as suggested (lines 78-92 in the revised manuscript).
4. “The conclusion could benefit from describing some tools needed to advance this field. One of the obvious things comes to mind in a large set of matched primary and metastatic tumors, both to lung and elsewhere. This type of tool would allow for answering the authors’ points of biomarkers that are lung-exclusive.”
This is a very insightful suggestion that compliments the contents of this review very well, and we have now added this to the revised manuscript (lines 393-397).